# Autistic Traits Related to the Importance of Interpersonal Touch and Appreciation of Observed Touch during COVID-19 Social Distancing

**DOI:** 10.3390/ijerph20186738

**Published:** 2023-09-11

**Authors:** Jutta R. de Jong, Hendrik Christiaan Dijkerman, Anouk Keizer

**Affiliations:** Experimental Psychology/Helmholtz Institute, Utrecht University, 3584 Utrecht, The Netherlands; c.dijkerman@uu.nl (H.C.D.); a.keizer@uu.nl (A.K.)

**Keywords:** autism, interpersonal touch, social distancing, autistic traits, touch observation, longing for touch, COVID-19

## Abstract

Studies have confirmed the significance of touch for psychological wellbeing. Social distancing regulations during the COVID-19 pandemic reduced people’s ability to engage in interpersonal touch and caused increased an appreciation for observed touch, as well as a longing for touch within the neurotypical population. Yet, while the impact of social distancing and the importance of touch are evident in neurotypical individuals, it remains unclear how these factors manifest in autistic individuals. Previous research has related high levels of autistic traits to reduced levels of perceived pleasantness of touch and a reduced interest in interpersonal touch. Our study aimed to examine the differences in the appreciation of observed touch and longing for touch during social distancing between individuals with low and high levels of autistic traits. We conducted an online survey on autistic traits, the appreciation of observed CT-optimal touch and longing for touch. Consistent with our predictions, our results confirmed that individuals with high levels of autistic traits evaluated videos depicting CT-optimal touch less favorably compared to those with lower scores on autistic traits. Additionally, only the group with low levels of autistic traits exhibited a longing for touch during social distancing, whereas the group with high levels of autistic traits did not. The results provide insights in the appreciation of touch in relation to autistic traits during the unique circumstances of the COVID-19 pandemic.

## 1. Introduction

As humans, we can appreciate differences when it comes to physical and social comfort. Within the general population, touch is an unmissable element of psychological well-being [1,2]. The significance of physical touch became even more apparent during the COVID-19 pandemic between 2020 and 2023 [3]. The restriction of the amount of received and given touches, which was evident during social distancing regulations within the Netherlands, resulted in an overall increase in longing for interpersonal touch within the neurotypical population [4,5,6]. Prior to the pandemic, the research had already emphasized the significance of touch among neurotypical individuals, stating that the hedonic value of interpersonal touch plays an important role in the appraisal of the touch itself [7]. Interpersonal touch in the form of slow and gentle stroking on the skin is perceived as pleasant [8]. This slow stroking can activate unmyelinated CT fibers present on hairy skin and is referred to as CT-optimal or affective touch [9,10]. Especially, stroking velocities of 3 cm/s (best at 1–10 cm/s) result in the highest firing rates of CT fibers [10], whereas faster stroking (e.g., 18 cm/s or 30 cm/s) will not active these CT-fibers [11]. A relationship was found between stroking velocities and the perceived pleasantness of touch, with CT-optimal touch resulting in a higher subjective appraisal of touch compared to that of CT-non-optimal touch stroking velocities among neurotypical people [8,12]. Neuroimaging studies on the processing of touch have linked activation in insular areas (left, middle and posterior regions), the temporoparietal junction and the somatosensory cortex to the processing of touch. Moreover, these studies have specified unique activity in the medial prefrontal cortex and superior temporal sulcus (right posterior) in response to CT-optimal touch [12,13,14,15,16]. These latter brain areas have been associated with the perception of touch within a social context and theory of mind [17]. Furthermore, we know from the previous literature that the mere observation of touch can activate the primary somatosensory cortex [18]. The observation of CT-optimal touch was found to result in the activation of brain areas that were associated with tactile stimulation at a CT-optimal speed [19]. The observation of touch and perceived pleasantness were also studied during social distancing. A previous study [4] found that higher levels of longing for touch corresponded to increased pleasantness ratings of videos depicting interpersonal touch. In other words, the more participants longed for touch, the more they appreciated viewing interpersonal touch. Thus, the previous literature underlines the importance of interpersonal touch and describes neural pathways that are related to the experienced and observed pleasantness of touch. It became apparent that during social distancing, neurotypical people experienced higher levels of longing for touch and that the perceived pleasantness of observed touch was also affected. However, it is unclear if these findings can be translated to neurodiverse populations that have been found to have a different relationship with touch compared to that of neurotypical people.

The neurodiverse population, whom we know from previous work experience touch differently compared to neurotypical people, are individuals diagnosed with Autism Spectrum Disorder (ASD). ASD is defined as a neurodevelopmental disorder associated with atypical responses to sensory stimuli [20,21,22,23,24,25,26], distinct social interaction [27] and linked to lower levels of interpersonal touch frequency [28]. Autistic individuals have been found to show over-responsiveness in the form of hypersensitivity to tactile stimuli [21,23,29,30]. For example, previous research compared tactile sensitivity between autistic and non-autistic groups and found that the autistic participants experienced their own touch as well as touch of others as more intense [31]. This study highlighted tactile hypersensitivity in autistic individuals, while also demonstrating that their own touch does not affect them in a different way compared to neurotypical people. Others reported tactile hypersensitivity in the form of pulling away when touched, stiffening when touched [22,32] and having less tolerance to hugs [25]. Although these previous studies have illustrated the differences between autistic individuals and neurotypical people based the overall attitude towards touch, it is important to note that most of the findings on the perceived pleasantness of touch have been derived solely from studies conducted within the neurotypical population. A previous study looked at different forms of touch and their relationship with autistic traits [28]. They introduced affective touch awareness as a measurement of the different ratings between CT-optimal touch and CT-non-optimal touch and found a negative relationship between autistic traits and affective touch awareness. In other words, higher autistic traits were related to a smaller difference in the appraisal of CT optimal and CT non-optimal touches. In addition, neuroimaging studies that focused on the hedonic value of touch reported that compared to non-autistic individuals, brain regions associated with CT-optimal touch were less responsive in autistic individuals during tactile stimulation at a CT-optimal speed. Their studies reported no differences in brain activity in response to CT-optimal touch and CT-non-optimal touch within an autistic group [33,34]. Moreover, others reported a negative relationship in brain activity between autistic traits and the areas associated with CT-optimal touch [35]. Furthermore, previous research [28] reported a connection between lower levels of interpersonal touch frequency and ASD. Lastly, a difference between autistic and non-autistic groups was reported based on affective touch awareness, but established no relationship between the amount of autistic traits and affective touch awareness [36]. Taken together, the previous literature describes lower levels of interpersonal touch frequency [28] and indicates less appreciation for touch in relation to either autistic individuals [33,34,36] or those with autistic traits [28,35].

Social distancing regulations during the COVID-19 pandemic created a unique opportunity to study individual touch preferences in relation to autistic traits during reduced levels of interpersonal touch within a large sample of the population. Recent studies during social distancing highlighted the importance of touch by reporting high levels of longing for touch [4,5,6], which even manifested in increased experienced pleasantness when merely observing touch [4], whereas prior studies on ASD and autistic traits dating to before social distancing rules were enforced describe a reduced overall interest in and perceived pleasantness of touch [28,33,34,35,36]. Therefore, in the current study, we investigated whether the perceived pleasantness of observed touch and the amount of longing for touch differed between individuals who scored low and high on autistic traits during a period of social distancing. Based on previous findings that showed differences based on autistic traits [28,35] and our aim to include a wide age span (a previous study [36] described their narrow age span as a limitation), we chose to study autistic traits within the general population and link these traits to the experience of touch, regardless of whether a formal ASD diagnosis was established. Our study aim was twofold. First, we studied the pleasantness ratings of observed CT-optimal touch as well as affective touch awareness and differences based autistic traits. We expected higher pleasantness ratings for observed CT-optimal touch and affective touch awareness among individuals who scored low on autistic traits, as compared to those who scored high on autistic traits. These expectations were based on the fact that high levels of pleasantness were found for observed CT-optimal touch during [4] and before social distancing [19,37] within the neurotypical population. In addition, previous studies describe a relationship between lower levels of affective touch awareness at the behavioral level [28] and lower neural responses to CT-optimal touch in relation to higher levels of autistic traits [35]. We included the observation of CT-non-optimal touch as a baseline against CT-optimal touch in order to calculate affective touch awareness.

Second, we focused on the differences in longing for touch between individuals who scored either high or low on autistic traits by asking participants to report their interpersonal touch frequency during social distancing. We expected to find lower levels of longing for touch among individuals who scored high on autistic traits compared to individuals who scored low on autistic traits during social distancing, since autistic individuals have previously been linked to lower levels of interpersonal touch frequency [28] as well as atypical responses to touch in general [20,21,22]. We speculated that due to the less prominent role of interpersonal touch, social distancing would not co-occur with longing for touch when they scored high on autistic traits. Although we only acquired data during COVID-19, we aimed to capture the appreciation of interpersonal touch frequency before social distancing by including a retrospective self-report question. For the latter part, we did not expect to find differences based on autistic traits prior to the outbreak of COVID-19.

## 2. Materials and Methods

### 2.1. Participants

Between 25 June and 25 July 2021, data were collected through an online questionnaire on Qualtrics. At the time of the study, the Netherlands was in a national lockdown [3]. This entailed that the government advised all citizens to stay at home as much as possible, keep 1.5 m distance from others, wear a face mask in public places and to cancel all social gatherings and interactions [3]. For this study, we recruited a community sample consisting of Dutch residents over 16 years of age. In total, 707 participants participated in the study. The participants who completed less than 75% of the experiment were excluded, as well as two participants with anomalous data who showed up as outliers in 2 variables. This resulted in a final sample of 377 for analysis. The majority of these participants were female (*N* = 237) and aged 16–66 (*M* = 32.37, *SD* = 11.04).

### 2.2. Materials

#### 2.2.1. Autism Questionnaire

As a self-report measure of autistic traits, the AQ-short was used [38], which is based on the Autism Spectrum Quotient [39], containing 28 questions concerning social skills, routine, switching and imagination. Scores ranged from 1 (*strongly disagree*) to 5 (*strongly agree*), and the AQ-short included questions such as “*If there is an interruption, I can switch back very quickly*”, “*I tend to notice details that others do not*” and “*New situations make me anxious*”. A higher total score on the AQ-short corresponds with higher levels of autistic traits. Cronbach’s alpha in the current sample was high (α = 0.904). Bases on quartiles, the groups were selected. The low AQ group contained participants from Q1 (*N* = 95, with a range of 35–58 AQ scores), and the high AQ group contained participants from Q4 (*N* = 90, with a range of 82–107 AQ scores). A complete overview of demographic characteristics for both groups can be found in Table 1. 

#### 2.2.2. Interpersonal Touch Observation

The pleasantness of observed CT-optimal touch was measured by presenting two videos that depicted someone stroking the forearm of another person. We used the same stimuli as in a previous study on pleasantness of observed touch [3], where the CT-optimal touch video showed a hand slowly stroking an arm at a 3 cm/s velocity. The CT non-optimal videos showed faster stroking velocities of the hand on the arm (30 cm/s) (see Figure 1). The videos were counterbalanced across participants and had a duration of 10 s. The participants were instructed to watch the videos and answer questions afterwards. After each video, the participants completed a short questionnaire in which they rated the video. The participants answered four questions regarding the appreciation of the touch: “*How did the video make you feel?*”, “*What did you think of the touch in the video?*”, “*How did you experience the video?*” and “*How much would you like to be touched in this way?*”. The questions were rated on a 10-point scale ranging from either “*Very unplea**s**sant*” to “*Very pleasant*” to the portrayed touch or from “*Not at all*” to “*Very much*” in the context of being touched. After the questionnaire was completed, a mean score for touch appreciation was calculated, where a higher score indicated that the observed touch in the video was more appreciated in terms of pleasantness and wishing to be touched as portrayed in the videos. Scores of 50 would reflect a neutral appreciation of the portrayed touch. Last, we calculated affective touch awareness by subtracting CT-non-optimal touch from CT-optimal touch based on [28,36].

#### 2.2.3. Longing for Touch

We assessed touch appreciation before social distancing was enforced by asking the participants to rate the following statements on a VAS scale ranging from 0 (*not enough*) to 100 (*enough*): “*Before the social distancing period I felt I was touched…*”. We assessed longing for touch with two self-report questions, namely: “*Currently I would prefer to be touched by others” and “Currently I would prefer to touch others*”, which is in line with [3]. Scores could range from 0 (*less preferred touch*) to 100 (*more preferred touch*), where a score of 50 would reflect a perfect balance between the wish to be touched and actually receive a touch, and scores > 50 would reflect a longing for touch. The mean score of the total scores of two questions was used to indicate longing for touch. Cronbach’s alpha in the current sample of 2 items was α = 0.902. Additionally, we included a questionnaire to measure the amount of longing for touch [40] based on different levels of touch from family members and strangers. However, due to methodological issues, we did not include this questionnaire in the analyses.

## 3. Results

### 3.1. Data Analysis

The normality of all the scores was assessed. The Shapiro–Wilk test indicated that all the outcome variables, except for affective touch awareness within the low AQ group, *W* (95) = 0.976 and *p* = 0.085, were not normally distributed. Therefore, we decided to conduct non-parametric tests. An overview of outcome scores based on all the AQ scores can be found in the Appendix A.

### 3.2. Demographics

A Chi-Square Test of Independence was performed to assess the differences between groups based on demographic variables using Bonferroni-adjusted alpha levels of 0.01 per test (0.05/5). We found no significant difference between groups for age*: X**^2^*** (1, *N* = 157) = 38.92, *p* = 0.648; gender: *X**^2^*** (1, *N* = 185) = 2.64, *p* = 0.450; COVID-19 infections: *X**^2^*** (1, *N* = 185) = 2.06, *p* = 0.560; and relationship status: *X**^2^*** (1, *N* = 185) = 4.44, *p* = 0.035. However, we did find a significant difference for living situation *X**^2^*** (1, *N* = 185) = 10.43, *p* > 0.001, where participants in the low AQ group more often lived with others than the participants in the high AQ group did.

### 3.3. Pleasantness of Observed Touch

Our main question concerned the differences in pleasantness ratings of CT-optimal touch between the low and high AQ groups. The mean CT-optimal touch pleasantness rating for the low AQ group was *M* = 57.60 (*SD* = 25.43), and it was *M* = 30.11 (*SD* = 26.14) for the high AQ group. In line with our expectations, a Mann–Whitney U test showed significant differences between the low and high AQ group for pleasantness ratings of observed CT-optimal touch: *U* = 1925.00 *p* < 0.001. We included CT-non-optimal touch as a control condition. The mean CT-non-optimal touch score for the low AQ group was *M* = 24.14 (*SD* = 16.14), and it was *M* = 13.74 (*SD* = 13.99) for the high AQ group. Mann–Whitney U test showed significant difference between the low and high AQ groups for CT-non-optimal touch: U = 2753, *p* < 0.001 Following previous studies [28,36], we calculated affective touch awareness by subtracting the pleasantness ratings of CT-non-optimal touch from the pleasantness ratings of CT-optimal touch. The mean affective touch awareness score was *M* = 34.46 (*SD* = 25.12) for the low AQ group, and it was *M* = 16.37 (*SD* = 24.92) for the high AQ group. A Mann–Whitney U test showed significant differences between the low and high AQ groups for affective touch awareness: *U* = 2505.00, *p* < 0.001, which was in line with our expectation that the low-level autistic trait group would show higher levels of affective touch awareness compared to those of the high autistic trait group, for CT-optimal touch. Taken together, the individuals in the high AQ group rated observed CT-optimal and CT-non-optimal touches as less pleasant than the low AQ group did, indicating that the high AQ group rated watching any form of touch as less pleasant than the low AQ group did. Moreover, the high AQ group had a significantly lower CT-optimal touch awareness score than the low AQ group did. This implies that the difference in pleasantness ratings between CT-optimal and CT non-optimal touches is larger in the low AQ group than it is in the high AQ group. These findings are visualized in Figure 2.

## 4. Longing for Touch

The mean longing-for-touch score for the low AQ group was 69.74 (*SD* = 26.43), and it was 36.93 (*SD* = 25.97) for the high AQ group. Scores above 50 indicate a longing for touch. A Mann–Whitney U test showed a significant difference between the low and high AQ groups for longing for touch during social distancing, *U* = 1570.50, *p* < 0.001. In line with our hypothesis, the low AQ group showed a longing for touch, whereas the high AQ group did not. In addition, we explored the appreciation of the interpersonal touch frequency before social distancing for the low AQ group (*M* = 75.60, *SD* = 24.67) and high AQ group (*M* = 78.52, *SD* = 26.47). A Mann–Whitney U test showed no differences between the low and high AQ groups, *U* = 3806.50 *p* = 0.189, meaning that individuals within the high and low AQ groups were in retrospect equally satisfied with their interpersonal touch frequency before social distancing. Taken together, these results suggest that, looking back, both groups were equally content with how often they experienced touch prior to the pandemic. However, we found that during the pandemic, the low AQ group experienced a longing for touch whereas the high AQ group did not.

## 5. Discussion

In our study, we aimed to investigate the differences in the perceived pleasantness of touch and longing for touch based on autistic traits during the exceptional circumstances of social distancing during the course of the COVID-19 pandemic. The importance of interpersonal touch and its beneficial effects on psychological wellbeing have been established in multiple studies [1,2]. Social distancing restrictions during the COVID-19 pandemic severely restricted people’s ability to engage in interpersonal touch, which has been linked to changes in the perceived pleasantness of observed touch [4], as well as increased levels of longing for touch [4,5,6]. However, although the link between social distancing and the importance of touch has been repeatedly made within the neurotypical population [4,5,6], it is unclear if this would be the case in relation to ASD. Autistic individuals, as well as those with autistic traits, were previously linked to an altered experience of pleasantness with respect to touch [28,33,34,35,36]. To this day, no previous study has focused on autistic traits measured by AQ scores and touch perception during COVID-19 social distancing. This includes measuring affective touch awareness based on virtually observed touch. We aimed to shed light on these topics by comparing groups with low and high levels of autistic traits within a community sample. Previous research [38] reported AQ scores of 87.76 (*SD* = 12.06) for autistic males and 91.49 (*SD* = 11.62) for autistic females, which were similar to our mean AQ score in the high AQ group (M *=* 90.16 (*SD* = 6.21). In addition, our low AQ group was similar to the non-autistic control group from this research [38]. The results from this previous research [38] included two control groups; A Dutch non-autistic control reference contained mean scores of 56.91 (*SD* = 9.32) for males and 52.79 (*SD* = 8.06) for females, and a Dutch non-autistic control replication had ranges of 58.40 (*SD* = 8.70) for males and 56.61 (SD = 8.63) for females. Since our low AQ group mean score was 49.71 (*SD* = 6.09), this is roughly in line with the clinical population included in a previous study [38].

In order to assess the differences based on autistic traits, our first aim was to study the perceived pleasantness of observed Ct-optimal touch and compare high and low AQ groups based on their pleasantness scores. We expected that, for the low AQ group, the touch pleasantness ratings for videos depicting CT-optimal touch would be higher compared to those in the high AQ group. This hypothesis was confirmed by our results; we indeed found individuals in the high AQ group who appraised videos depicting CT-optimal touch with lower scores compared to those in the low AQ group. Our findings were in line with the previous literature on CT-optimal touch, indicating that, within the neurotypical population, this form of touch was perceived as pleasant [8,9,10,11,12,13,14], while this was not reported in relation to ASD [28,33,34,35,36]. Furthermore, we calculated touch awareness scores based on different ratings between CT-optimal touch and CT-non-optimal touch. We found that the touch awareness scores within the low AQ group were higher compared to those of the high AQ group. This indicates that, for the high AQ group, the differences in perceived pleasantness between the observed CT-optimal and CT-non optimal touches were smaller compared to those in the low AQ group. Our findings were in line with a previous study, which reported lower levels of affective touch awareness in relation to autistic traits [28]. Taken together, this study was able to relate the pleasantness of observed touch to autistic traits during the global COVID-19 pandemic. For the first time, our study demonstrated that high-level autistic traits can be linked to lower pleasantness levels of observed touch and smaller differences between observed CT-optimal and CT-non optimal touches. This further expands our knowledge on observed touch in relation to autistic traits within a community sample. A relevant sidenote when discussing observed touch is the interpretation of the differences between stroking velocities for the observed CT-optimal and CT-non-optimal touch conditions. Previous studies have reported altered visual motion perception for autistic individuals when comparing to that of the controls [41,42] and discussed the importance of biological motion and social perception [43]. In future work, it would be interesting to assess visual motion disturbances and see if these can be related to the perception of observed interpersonal touch depicting different stroking velocities. 

Our second aim was to study autistic traits and the presence of a longing for touch, as well as the reported satisfaction with the interpersonal touch frequency before social distancing. In line with our expectations, the low AQ group showed a longing for touch during social distancing. More importantly, we did not find a longing for touch by individuals within the high AQ group. Previous studies found a longing for touch within the general population during social distancing [4,5,6]. Moreover, autistic individuals were previously linked to decreased levels of interpersonal touch frequency [28], as well as hypersensitivity to tactile stimuli [21,23,29,30] and averse responses to touches [22,25,32]. Lastly, we observed that before the COVID-19 pandemic, individuals who scored either high or low on autistic traits tests did not seem to differ in their reported appreciation for the amount of touch they received. Our current findings were able to shed more light on their satisfaction of interpersonal touch and its relationship with autistic traits. Previous studies thus far were able to highlight the differences between neurotypical people and autistic individuals regarding touch [21,22,23,25,28,30,32]. Our study had the unique focus of linking autistic traits with satisfaction towards interpersonal touch frequency during the COVID-19 pandemic. Interestingly, we found that only the individuals who scored low on autistic traits reported a longing for touch during social distancing. For the first time, our results demonstrate that during a period of social restrictions with respect to touch, individuals who score high on an autistic traits test report to experience no longing for touch. Based on a self-report on the satisfaction of interpersonal touch frequency before social distancing, we did not find a difference based on autistic traits. However, this does not imply that the amount of touch the individuals received before social distancing was similar. A previous study already linked ASD to decreased levels of interpersonal touch frequency [28]. Thus, it is possible that individuals in the high AQ group would have received fewer touches before social distancing compared to those of the low AQ group and are equally satisfied with this amount. If so, the relative difference in the amount of touches during social distancing may have been smaller in the high AQ group compared to the low AQ group, which may explain the reduced longing for touch. In future research, it would be interesting to replicate our findings by including an extra condition of physical touch in order to see if there are differences between observed and physical touches based on autistic traits. In addition, including factors such as mental well-being would be worthwhile, since touch is important for people’s well-being [1,2] and a longing for touch during COVID-19 was linked to an increase in feelings of loneliness and anxiety [6]. Moreover, it would be interesting to study interpersonal touch frequency differences within a group of individuals who received an ASD diagnosis. Furthermore, future studies on either ASD or autistic traits could compare interpersonal touch preferences based on the familiarity of the toucher [40] or the overall experience of touches during their life [44]. We found that participants in the low AQ group reported to be living with others more frequently compared to those in the high AQ group did. It would be interesting to further explore the differences based on demographic variables in relation to either ASD or autistic traits, and other studies could focus on which confounding factors have an influence on the experience of longing for touch. A limitation of our study was that due to the sudden onset of the COVID-19 pandemic, it was impossible to gather data on touch satisfaction before the social distancing regulations were enforced. We therefore had no other option than to include a retrospective self-report measure, which can be less reliable and should be interpreted with a critical eye. However, the majority of our measurements during these unique circumstances do provide a solid base for future research on ASD and touch to compare touch appreciation after social distancing regulations. It would be interesting to replicate our results between a restricted society and an unrestricted society in order to determine if these differences based on autistic traits were an effect of social distancing or not.

## 6. Conclusions

We show the differences in appreciation of observed touch as well as longing for touch based on autistic traits. We illustrated that touch appraisal differences were not limited to actual touch, but they are evident during the mere observation of touch. Moreover, we demonstrated that during social distancing regulations, only those scoring low on an autistic traits test experienced a longing for touch. Our study contributes to the understanding of how the value of touch can be related to autistic traits. This provides an important stepping stone for future studies that aim to study the appreciation of touch in relation to autistic traits and ASD. We further demonstrated the feasibility of studying autistic traits within a community sample, presenting broader possibilities for studying ASD and its association with touch. Furthermore, we have extended the existing body of knowledge by demonstrating that touch can be studied through the presentation of visual depictions of tactile experiences. Moreover, we illustrated that there has been significant diversity during COVID-19 regarding longing for touch, which was not universally high across the entire population. Lastly, our study contributes to how different individuals experienced the COVID-19 social distancing period in terms of preferred interpersonal touch.

## Figures and Tables

**Figure 1 ijerph-20-06738-f001:**
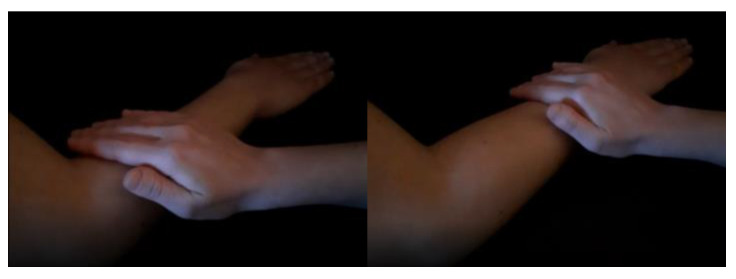
Participants were presented with videos depicting either CT-optimal touch or CT-non-optimal touch.

**Figure 2 ijerph-20-06738-f002:**
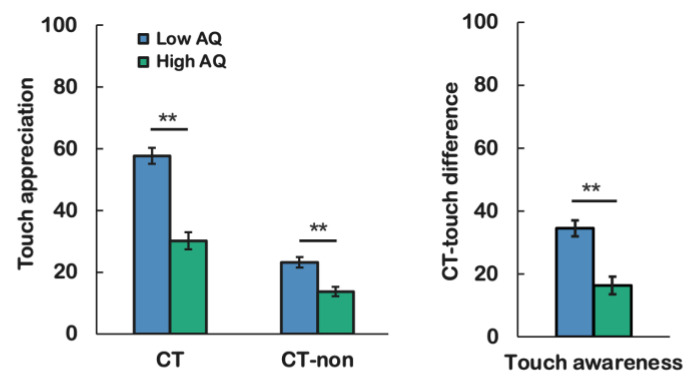
Pleasantness of observed CT-optimal and CT-non-optimal touch (**left**), as well as affective touch awareness for both low and high AQ groups (**right**). Error bars represent standard error mean (SEM) and *p < 0.001* was flagged with 2 stars (**).

**Table 1 ijerph-20-06738-t001:** Demographics of AQ groups.

	Low AQ		High AQ	
	N	%	N	%
**Gender**				
Male	35	36.84	28	31.11
Female	60	63.16	60	66.67
Intersex	0	0	1	1.11
Rather not say	0	0	1	1.11
**COVID-19**				
I am currently infected	2	2.11	0	0
I was previously infected	10	10.53	9	10
I am not/never was infected	68	71.58	68	75.56
Unsure	15	15.79	13	14.44
**Romantic partner**				
Yes	59	62.11	48	53.33
No	36	37.89	48	53.33
**Living with others**				
Yes	74	77.89	50	55.56
No	21	22.11	40	44.44

## Data Availability

The data presented in this study are available on request from the corresponding author.

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
