# Peer review of "Autistic Traits Related to the Importance of Interpersonal Touch and Appreciation of Observed Touch during COVID-19 Social Distancing"

_ijerph, 2023, doi:10.3390/ijerph20186738_

Round 1

Reviewer 1 Report

Thank you for the opportunity to review an interesting manuscript.

Overall, the topic is very relevant for mental health discussions and to improve our understanding of social touch in autism in relation to special circumstances as social distancing during the COVID-19 pandemic.

As a first comment, a note of caution on the direct generalization of findings to the autistic population. Although level of AQ scores were in line with previous findings on autistic participants for the high AQ group, the AQ alone cannot be considered equal to a clinical diagnosis of autism. The authors should highlight this and interpret their findings in relation to high level of autistic traits, requiring further replication in a clinical sample.

The background section describes previous MRI work and neurotypical/autistic parallels very nicely. Yet, I would like to raise the question on the interpretation of reduced social touch in autism, evident from early manifestations already, and here attributed to reduced interest (e.g., in line 31). Can we really be so specific in relation to interest? Or could the interpretation be more nuanced in relation to broader sensory features of autism, as hyper/hypo-responsivity? 

About demographics, given the high chronogeneity of sensory features in autism, was there any effect of age given the wide range in the study sample?

The sex distribution was skewed towards females, while it is usually the contrary for clinical autistic samples. This requires further caution in the generalization of findings to the clinical autism category. The authors should discuss interpretation and generalizability of findings in relation to this unbalanced distribution.

Frequency of social touch is a relevant variable in the present study, as highlighted by the authors in the discussion of potential baseline differences. While there seems to be no data available on frequency of touch behaviours before the pandemic, data showed relevant differences in romantic relationships (which might be used as proxy for previous touch experience) and living situation (which might inform on potential differences in current social touch experience) between groups. Did the authors check for potential confounding effect on the observed group differences?

The discussion of findings might benefit from the integration of known sensory features associated with the diagnosis or high level of autistic traits. For example, visual motion perception is known to be altered in autism, which might influence differential perception of optimal vs. non-optimal conditions. Similarly, perceived pleasantness of touch from visual stimuli might be altered in the presence of synaesthesia, which is often a concurrent condition in autism.

Also, ecological validity of stimuli should be considered, with previous work in autism showing relevant differences in perceptual experience of autistic participants between image/video stimuli and psychophysical/live experiments.

Overall, this study has the potential to inform us on the influence of social distancing measures on mental well-being across the population based on stratification for autistic traits and in relation to touch. It would be extremely valuable if the authors had the opportunity to actually test this (which of course depends on data availability, so I completely understand when this is not possible), or at least discuss this perspective in relation to previous work on differential effects of social distancing in relation to autistic traits.

Author Response

Attached our letter with detailed responses.

Reviewer 2 Report

The manuscript looks at touch preference in a convenience sample.

The study is partly a replication (video stimuli) and improvement of previous studies on the relation between autistic traits and touch preferences. The link to the pandemic though is thin and partly confusing in the other wise well written paper.

I suggest to structure clearer what is due to the pandemic (less longing for touch) and what is generally due (CT optimal vs CT non-optimal) and autistic traits.

One could first present the CT (video) results, highlighting the replication and proof of principle. Then report the results from the questionnaires (2-items) and retrospective, showing that persons with many autistic traits have less longing for touch.

regarding analysis, for plotting one can use the extreme groups (low and high autistic traits) but for analysis it is fine to use all 377 participants and their AQ short score (robust regression or Kendall's tau for rank correlations between CT optimal and AQ score and so on).

Did you look at the subscales of the AQ28? It would be interesting whether the CT touch preference score (and your other outcome measures) relate to social skills, routine, switching or imagination.

Please be also careful to use the terms persons with an autism spectrum diagnosis. Since you have no verification of a diagnosis (nor seem to have ask for it) your manuscript is about a personality trait, not about a clinical patient vs non-patient group. 

minor issues

line 89 remove "to"

line 123 [33] needs to be cited as author, year

line 233ff please report exact p values (if not p < .001).

Fig 3 would be more informative if it would be boxplots

from line 401 please update with your information (kept the default from the template)

table 1 could include the test statistics in a 5th column

none

Author Response

(The authors gave the same response as above.)

Round 2

Reviewer 1 Report

The authors addressed all the points raised appropriately. I have no further comments.